# Research on Cold Chain Logistics Transportation Scheme under Complex Conditional Constraints

**Bin Xu \*, Jie Sun, Zhiming Zhang**  **and Rui Gu**

School of Shipping Economics and Management, Dalian Maritime University, Dalian 116026, China; 1120210498@dlmu.edu.cn (J.S.); 1120210504zzm@dlmu.edu.cn (Z.Z.); gurui@dlmu.edu.cn (R.G.)
\* Correspondence: xb007@dlmu.edu.cn

**Abstract:** A mathematical model is proposed to minimize the sum of vehicle fixed cost, fuel cost, carbon-emission cost, cooling cost, time-penalty cost and split-compensation cost, on the basis of considering the three-level cold-chain-logistics network of manufacturer, distribution center, and seller. The model is constructed based on the constraints of customer time window, vehicle load, demand-splitable, and semi-open driving of multiple distribution centers. We to divide the customer areas according to geographical locations and to carry out the transportation processes in stages. The target solution, which includes vehicle routing, service time and type, cargo details, etc., has been formulated. A two-stage hybrid-heuristic-path-scheme solution algorithm that combines a taboo table, a genetic algorithm, an optimal-path-generation algorithm, a load-capacity-constraint algorithm, and a time-window-constraint algorithm is designed in view of the complexity of the model and the uniqueness of the solution scheme. This paper aims to reasonably plan the resource allocation of cold chain logistics enterprises, reduce the comprehensive cost of cold chain transportation, improve customer satisfaction, and respond to the green logistics policy advocated by the state by reducing vehicle transit time and fuel consumption, and promote energy conservation and emission reduction.

**Keywords:** cold chain logistics; demands splitting; time window; vehicle routing; cargo plan

## 1. Introduction

### 1.1. Importance and Motivation

$\quad$ With the improvement of living standards in developing countries, especially in China, consumer demand for fresh refrigerated and frozen food has increased significantly, thus promoting the rapid development of the cold chain industry. China's cold chain logistics market has reached RMB 1.437 billion yuan in 2022, and it was reported that the global cold chain logistics market will reach 5661.896 billion yuan by 2028, with a Compound Annual Growth Rate (CAGR) of 14.37% during the forecast period [1]. Distribution is an important link in cold-chain logistics, and it accounts for over 80% of the time required for cold-chain products to be transferred from the manufacturers to the final consumers [2]. Cold chain logistics distribution involves the delivery of perishable food, agricultural products, or drugs to retailers or supermarkets in different locations with minimal transportation time and cost. As an important part of cold chain logistics, the distribution process affects not only the customer service levels, but also the logistics operating costs and the quality of cold chain products [3]. Macharis et al. [4] pointed out that urban goods distribution (UGD) has an important impact on the sustainable development of cities. According to a report, the economic losses due to the lack of efficient path design and refrigeration during cold chain transportation even exceeded one hundred billion RMB just for vegetables and fruit in China from 2012 to 2015, let alone for other agricultural products [5]. Therefore, there is an urgent need to optimize the routing path of the vehicles in cold chain logistics distribution considering the freshness of perishable agricultural products as well as total cost. The routing of cold chain logistics is a special application in the vehicular routing problem

(VRP), which is more concerned with the delivery time of perishable products [3]. However, the development of cold chain logistics still has problems, such as high distribution cost, the high rate of cargo loss, and serious environmental pollution. Therefore, it is of great importance to rationalize the dispatch of vehicles in the process of cold chain logistics in order to achieve significant enterprise and social benefits.

### 1.2. Literature Review

In the existing research on cold chain logistics transportation optimization, most scholars' research focuses on cold chain logistics distribution path optimization. With the improvement of living standards, consumer demand for fresh, refrigerated, and frozen foods is gradually increasing, thus promoting the rapid development of the cold chain industry. Most scholars in domestic and international research on cold chain logistics network optimization focus on optimizing distribution routes and selecting distribution centers. When it comes to optimizing distribution routes, the focus is on considering temperature constraints, minimizing distribution costs, and minimizing green costs. Among them is distribution route optimization. Sobhi et al. addressed the issue of perishable product freshness in cold chain transportation and developed a distribution path optimization model [6]. Chen et al. [7] analyzed the strict requirements of fresh agricultural products on the distribution process in cold chain logistics from the perspective of the low-carbon economy. They constructed a cold chain logistics vehicle routing optimization model that comprehensively considers the relationship between economic benefits and environmental impacts. They proposed a swarm intelligence optimization algorithm–particle swarm algorithm, which was improved in terms of the inertia weight, convergence factor, learning factor, and population size. They demonstrated the superiority of their improved algorithm by comparing it with the traditional particle swarm algorithm. Zou et al. [8] studied the three-level cold chain network of "producer-distributor-seller" and constructed a three-level cold chain integrated inventory model to maximize total profit. They used an adaptive genetic algorithm to solve the problem. Zhao et al. [9] proposed a cold chain logistics path optimization method by improving the multi-objective ant colony algorithm, aiming to minimize total cost, total time, and total carbon emissions. Fu et al. [10] proposed a fresh product cold chain logistics delivery path optimization method based on a hybrid genetic algorithm, considering factors such as energy consumption and transportation costs, in order to minimize total cost and total transportation time. Ren et al. [11] studied the cold chain logistics path optimization problem under multiple distribution centers and considered the factor of carbon emissions. The research results show that using optimization algorithms can reduce total costs and total carbon emissions. Xiong [12] proposed an ant colony algorithm-based optimization algorithm for cold chain logistics distribution paths, aiming to minimize total distance and transportation costs while ensuring that product temperatures remain within a specified range. Liu [13] designed a dynamic programming model based on the Meme algorithm for optimizing multi-objective cold chain logistics deployment paths. The model aims to minimize transportation costs and transportation time while considering product temperature restrictions. Pan [14] proposed a conventional cold chain logistics distribution path optimization model from a low-carbon perspective for agricultural products. The model considers factors such as logistics costs, energy consumption, and carbon emissions, aiming to minimize total costs and total carbon emissions. Chen [15] proposed a green cold chain logistics location and path optimization method based on an improved genetic algorithm from a low-carbon and environmentally friendly perspective. The method aims to minimize total costs and total carbon emissions. Liu et al. [16] studied the optimization of cold chain logistics transportation routes to achieve energy conservation and emission reduction goals. The research conclusion shows that using optimization algorithms can reduce transportation costs and carbon emissions. Zhang [17] proposed a fresh cold chain logistics distribution path optimization method based on a genetic algorithm. The method considers factors such as transportation costs, energy consumption, and time, aiming to minimize total costs and total transportation time.

Wu et al. [18] proposed a comprehensive cold chain vehicle path optimization model to minimize the unit cost of product freshness and a carbon trading mechanism to calculate the cost of carbon emissions. The model considers cost, product freshness, and carbon emission environmental factors simultaneously. An improved adaptive chaotic ant colony algorithm is used to compute experiments on the model, and the feasibility and effectiveness of the model are analyzed by classical examples and practical cases. Wu et al. [19] proposed a low-carbon fresh food cold chain logistics distribution route optimization model considering customer satisfaction, and combined time, space, weight, distribution rules, and other constraints to optimize the distribution model. In this paper, the improved A* algorithm and ant colony algorithm are used to construct the model solution. The effectiveness, efficiency, and correctness of the single-objective low-carbon fresh produce cold chain model designed with the improved ant colony algorithm are verified through the simulation analysis results of different computational arithmetic cases. Kang et al. [20] proposed a logistics distribution route optimization model with total cost minimization as the objective function under the carbon tax system. The model integrates the technical advantages of IoT and the characteristics of cold chain logistics; introduces soft time window, customer satisfaction, and carbon emission as the main constraints; solves the mathematical model using an improved genetic algorithm; uses a matrix for coding; and demonstrates the effectiveness and rationality of the model and algorithm through examples. Zhou et al. [21] proposed to use of green technology to solve a variety of fresh produce cold chain distribution path optimization problems with transportation, refrigeration, and carbon emission cost as the objective function. A hybrid particle swarm optimization (HPSO) algorithm was designed to solve the problem of minimum freshness requirements for different types of cold chain distribution. Ma et al. [22] optimized local distribution paths for immediate demand, balancing enterprise economic efficiency and customer satisfaction while reducing the environmental pollution. To minimize distribution costs and maximize customer satisfaction, we designed an improved ant colony algorithm to solve the initial distribution path with an insertion method for immediate customer demand. The results show that the proposed model and algorithm are practical in meeting the sustainable development of cold chain logistics in China. Li et al. [23] proposed a multi-objective CCL model with the objectives of minimal carbon transaction cost, minimal network cost, and maximum customer satisfaction. Experimental data showed that the proposed method effectively improved customer satisfaction, reduced the total distribution cost, and promoted energy saving and emission reduction. Li et al. [24] proposed an optimized fresh food distribution path model with the minimum total cost and carbon emission as the objective function, and used an improved ant colony algorithm to optimize and determine the optimal distribution path. The optimization results provide a basis for fresh food distribution in the capital sub-center. Dong [25] proposed a TDVRP mathematical model for the low-carbon distribution of fresh agricultural products in the cold chain, taking the sum of the vehicle transportation cost, the cargo damage cost of fresh agricultural products, and the refrigeration cost of fresh agricultural products as the economic cost of distribution, and carbon emission cost as the environmental cost, in order to minimize the sum of economic cost and the environmental cost of distribution and customer satisfaction, and time penalty cost as the constraints. Qin et al. [26] proposed a comprehensive cold chain vehicle path optimization model with the minimization of customer cost per unit of satisfaction as the objective function. The cost of the cold chain logistics path optimization problem, customer satisfaction, and the carbon emission cost were also considered. The model enriches the optimization study of cold chain logistics distribution, and the research results complement the study of the effect of carbon price on carbon emissions and customer satisfaction. Liao et al. [27] proposed a green vehicle path problem (VRP) for perishable products to optimize the operation cost, deterioration cost, carbon emissions, and customer satisfaction. The model also considers time windows, different travel times during peak and off-peak hours, and working hours. The paper solves the proposed model using the multi-objective gradient evolution (MOGE) algorithm. Experimental results show that the MOGE algorithm has more significant

results than other algorithms. Goodarzian et al. [28] proposed a new responsive green cold vaccine supply chain network and designed a new mathematical model for the multi-objective, multi-cycle, multi-echelon distribution-distribution-positioning problem. The paper proposed Gray Wolf Optimization (GWO) and Variable Neighborhood Search (VNS) algorithms to solve the model, and the experimental results showed that the algorithm has higher quality and better performance than other algorithms. Fang et al. [29] transformed energy saving and emission reduction in the green cost into the path optimization problem and established a mathematical model of cold chain logistics path optimization with total cost minimization as the research objective. To address the problem of slow convergence due to insufficient pheromones in the initial stage of the ant colony algorithm, a hybrid ant colony algorithm was constructed by combining the A* algorithm with the ant colony algorithm, using the global convergence of the A* algorithm and the positive feedback of the ant colony algorithm. The effectiveness of the model and algorithm was verified through simulation optimization and the comparative analysis of the examples. Li [30] proposed a collaborative optimization model of urban fresh agricultural products' cold chain logistics inventory distribution based on distribution centers, proposed a partitioning solution strategy for the multi-distribution center problem, and proposed a collaborative optimization urban fresh agricultural products logistics inventory distribution system.

Jing [31] proposed an optimization model of demand-splitable cold storage multi-temperature co-distribution for the problems of high distribution cost and low vehicle loading rate in the traditional single-temperature distribution model. Chang [32] also improved the traditional single-temperature distribution model and proposed a multi-temperature co-distribution vehicle path optimization model, with significant results. Zhang [33] constructed a splitting mathematical model for multi-center semi-open delivery and pickup demand with the goal of the shortest vehicle delivery distance for the vehicle path problem. The experiments show that the multi-center semi-open delivery and pickup demand splitting model is better than the single-center delivery and pickup demand splitting model under independent delivery. At present, most convenience store chains use single-temperature cold chain vehicles for distribution, which a have serious duplication of vehicles and a low distribution efficiency. Chen [34] proposed an optimization model of demand splitting and cold storage multi-temperature co-distribution considering customer satisfaction, and compared the original distribution plan and the optimized distribution plan of the company. Li [35] proposed a multi-center semi-open cold chain logistics vehicle route optimization model considering carbon emissions, and showed that multi-center semi-open is more advantageous in cost reduction and route length reduction through the comparison of distribution modes. Jiang et al. [36] proposed a dual-objective cold chain logistics path optimization model for the multi-center semi-open vehicle path problem with time windows, with the objectives of minimizing total distribution costs and maximizing customer satisfaction. Fan [37] proposed a time-varying multi-temperature co-distribution path optimization model under the time-varying road network by considering the cargo mixing problem in cold chain logistics transportation, and found that the mixed load of goods in the same temperature layer can effectively improve the vehicle utilization rate and reduce the distribution cost by analyzing the solution results.

The vehicle loading scheme belongs to the category of the vehicle loading problem, which represents the relationship between goods and distribution vehicles, i.e., making full use of vehicle load under various constraints to achieve effective loading. Compared with the vehicle path problem, there are fewer studies on vehicle loading problems at home and abroad, and they mainly focus on the research direction of making full use of cargo loading space to improve space utilization. Paquay et al. [38] studied the multi-box size crating problem and constructed a mathematical model considering complex constraints such as stability, item fragility, box rotation, and item center of gravity distribution. Ding [39] proposed a split-loading strategy considering the characteristics of dispersed customers and long spacing at the end of rural "last mile" logistics and verified its superiority and rationality through practical cases. Wang [40] jointly considered the vehicle loading problem

and vehicle path problem, established a multi-objective optimization mathematical model, improved the existing crating algorithm, and proved the effectiveness of the algorithm through experiments. Furthermore, the problems of single-vehicle loading or multi-vehicle loading, single cargo loading, or multiple cargo loading are usually considered jointly with the vehicle path problem. Few scholars in China and abroad have studied them.

Table 1 compares some of the research results of the optimization of cold chain logistics solutions.

**Table 1.** Comparison of research results (Y: Yes, N: No).

| Literature | Conditional Constraints | | Cargo Loading Solutions | Routing Solutions | Two-Way Logistics |
|---|---|---|---|---|---|
| Wu et al. [19] | (Y)Time window | (Y)Weight capacity | N | Y | N |
| | (N)Mixed cargo loading | (N)Demand splitting | | | |
| | (N)Muti-distribution center semi-open | | | | |
| Zhou et al. [21] | (Y)Time window | (N)Weight capacity | N | Y | N |
| | (N)Mixed cargo loading | (N)Demand splitting | | | |
| | (N)Muti-distribution center semi-open | | | | |
| Dong [25] | (Y)Time window | (Y)Weight capacity | N | Y | Y |
| | (N)Mixed cargo loading | (N)Demand splitting | | | |
| | (N)Muti-distribution center semi-open | | | | |
| Jing [31] | (N)Time window | (Y)Weight capacity | N | Y | N |
| | (N)Mixed cargo loading | (Y)Demand splitting | | | |
| | (N)Muti-distribution center semi-open | | | | |
| Chang [32] | (Y)Time window | (N)Weight capacity | N | Y | N |
| | (N)Mixed cargo loading | (Y)Demand splitting | | | |
| | (N)Muti-distribution center semi-open | | | | |
| Zhang [33] | (N)Time window | (N)Weight capacity | N | Y | Y |
| | (N)Mixed cargo loading | (Y)Demand splitting | | | |
| | (Y)Muti-distribution center semi-open | | | | |
| Chen [34] | (Y)Time window | (Y)Weight capacity | N | Y | N |
| | (Y)Mixed cargo loading | (Y)Demand splitting | | | |
| | (N)Muti-distribution center semi-open | | | | |
| Jiang et al. [36] | (Y)Time window | (N)Weight capacity | N | Y | N |
| | (N)Mixed cargo loading | (N)Demand splitting | | | |
| | (Y)Muti-distribution center semi-open | | | | |
| Fan [37] | (Y)Time window | (N)Weight capacity | N | Y | N |
| | (Y)Mixed cargo loading | (N)Demand splitting | | | |
| | (N)Muti-distribution center semi-open | | | | |
| Paquay et al. [38] | (N)Time window | (Y)Weight capacity | Y | N | N |
| | (N)Mixed cargo loading | (N)Demand splitting | | | |
| | (N)Muti-distribution center semi-open | | | | |
| Ding [39] | (N)Time window | (Y)Weight capacity | Y | N | N |
| | (N)Mixed cargo loading | (Y)Demand splitting | | | |
| | (N)Muti-distribution center semi-open | | | | |
| This study | (Y)Time window | (Y)Weight capacity | Y | Y | Y |
| | (Y)Mixed cargo loading | (Y)Demand splitting | | | |
| | (Y)Muti-distribution center semi-open | | | | |

Although some scholars have conducted a significant amount of research on cold chain logistics transportation problems, there are additional limitations in the existing literature.

(1)　In the actual transportation process of cold chain logistics, there will be a variety of constraints such as time window, vehicle load, mixed cargo, demand that can be split, and semi-open multiple distribution centers. In addition, most of the literature only considers some of these constraints, and therefore cannot meet the complex requirements of enterprises in the actual transportation process.

(2)　In the research on cold chain logistics transportation optimization, most of the literature focuses on path optimization, and the research on vehicle loading schemes mainly focuses on making full use of cargo loading space to improve space utilization, and the joint optimization of vehicle loading scheme and driving path is relatively rare. It is not possible to provide specific transportation solutions for cold chain logistics enterprises, which is not of practical significance for them.

### 1.3. Objective and Research Contribution

The research objectives of this paper are to analyze the vehicle management cost, transportation energy cost, refrigeration cost, carbon emission cost, loading and unloading cost, etc. in the process of cold chain transportation, to minimize the total transportation cost considering the time window of customers and the actual distribution situation of cold chain logistics enterprises, and to develop a reasonable loading plan and transportation route.

In order to cope with the increase in the number of customers and the increase in the complexity of the model and algorithm due to the increase in the number of customers and the splitting of demand, we set up a mixed transport optimization model of "production-storage", "storage-sales" and "production-sales". In order to cope with the problems caused by the increase of customer types and demand splitting, and the increase of model and algorithm complexity, we establish a mixed transport optimization model of "production-warehouse, warehouse-sales, production-sales", and study a two-stage mixed heuristic path scheme optimization algorithm. The final solution can be directly applied in the actual transportation process of cold chain logistics enterprises.

The main contributions of this paper are as follows: (i) Theoretical implications: new solutions are provided for the cold chain logistics transportation optimization problem; (ii) Managerial implications: Establishing a cold chain logistics load scheme and route optimization model with the objective of minimizing the total cost of transportation, enriching the form of the solution, including both vehicle paths and load schemes, making full use of the existing resources of the enterprise, saving operational costs and maximizing efficiency. Additional implications are the centralized planning of enterprise resource allocation, considering the "supplier-seller" transportation process, and saving the inventory of enterprise cold storage. Shortening the time of vehicles in the distribution process to reduce the cost of damage, which is conducive to the improvement of customer satisfaction, fully demonstrates the enterprise's customer-centric business philosophy, while helping to enhance the core competitiveness of enterprises; (iii) Implications for the policymakers: By analyzing the actual operation of a cold chain transportation enterprise, this paper applies the theoretical research in real life, and responds to the green logistics policy advocated by the state by reducing the vehicle in-transit time and fuel consumption, which provides strong support for the enterprise's decision.

## 2. Problem Description and Model Assumptions

### 2.1. Problem Description

The research in this article focuses on the optimization problem of mixed transportation in scenarios involving "producer-warehouse, warehouse-customer, producer-customer". The problem is described as follows: A cold chain transportation company owns multiple distribution centers to serve multiple suppliers and sellers. The demands of each customer, their time windows, and their geographic locations are known. Homogeneous refrigerated trucks are used as the transportation tool, departing from any distribution

center, and returning to any center for replenishment during the journey, and are reusable. Customer orders include bulk goods and less-than-truckload (LTL) goods, with a variety of cargo types. Some cargo cannot be transported together. When the demand-splitting condition is met, orders from different customers can be split or combined for transportation. The goal is to minimize the total cost by arranging the vehicle loading plan and the driving routes.

### 2.2. Model Assumptions

The cold chain logistics cargo plan and route optimization process is a very complicated problem. In the process of mathematical modeling, many influencing factors and constraints are involved. This paper mainly considers the core constraints in the modeling process, and puts forward the following hypotheses at the same time conditions (suppliers and sellers are collectively referred to as customers):

(1) Assuming that there are multiple distribution centers with known locations and a sufficient supply of goods, each center has limited refrigerated vehicles to meet delivery demands, and the vehicle's carrying capacity is limited.

(2) Refrigerated vehicles depart from one distribution center, complete their demand, and can return to any distribution center. If time allows, the returning vehicles can perform deliveries again.

(3) The locations of each supplier, the amount of supplied goods, and the service time window are known. The locations of each seller, the demand for goods, and the service time window are known. There is no additional demand during the service process.

(4) Refrigerated vehicle service types can be divided into three categories: (a) directly delivering goods from the distribution center to the seller, (b) picking up goods from the supplier before delivering them to the seller, and (c) picking up goods from the supplier and returning to the distribution center for storage, with any combination of the three types under certain rule constraints. The refrigerated vehicle can pick up goods from suppliers and deliver them to sellers at any time.

(5) The distance between suppliers, distribution centers, and sellers are the actual road distance, and road traffic conditions are consistent, with refrigerated vehicles traveling at a steady speed during transportation.

(6) If the refrigerated vehicle arrives at the customer's location earlier than the service time window, it cannot provide the service in advance and incurs waiting costs. If it arrives later than the service time window, it can provide immediate service but incurs penalties. There is no fuel consumption or carbon emissions during waiting and service.

(7) Each vehicle can serve multiple customers, and each customer can be served by multiple vehicles.

(8) The divisibility of customer orders varies depending on factors that affect demand splitting, such as customer requirements, mixing relationships between cold chain products, and the operational feasibility of demand splitting. Each time the customer's demand is split, the customer pays additional compensation for the split. The amount of goods supplied by the supplier and demanded by the seller is less than or equal to the maximum carrying capacity of the refrigerated vehicle. If it exceeds the maximum carrying capacity, the customer must agree to split the demand.

### 2.3. Symbol Description

The known parameters of the cold chain logistics stowage scheme and path optimization model studied in this paper are shown in Table 2.

**Table 2.** Model symbol description.

| Symbol | Meaning |
|---|---|
| $Z$ | total delivery cost. |
| $M$ | Distribution Center Collection, $m \in M$. |
| $N$ | Supplier and seller demand node set, $n \in N$. |
| $U$ | The collection of demand points in a certain delivery of delivery vehicle $k$. |
| $K$ | The collection of delivery vehicles used, $k \in K$. |
| $Q$ | The maximum load capacity of the delivery vehicle. |
| $p_i$, $q_i$ | Pick-up quantity of supplier $i$, demand quantity of seller $i$. |
| $D_{ij}$ | The distance between nodes $i$ and $j$. |
| $f_k$ | The fixed cost of the kth delivery vehicle. |
| $\alpha_2$ | Cooling cost of a refrigerated truck per unit time during loading and unloading. |
| $\lambda_f$, $f_t$ | Fuel consumption rate of distribution vehicles (L/km). |
| $\lambda_c$, $c_t$ | Carbon emission rate of distribution vehicles (kg/km). |
| $t_i$ | Time required to load and unload goods at node $i$. |
| $\left[ ET_j,\ LT_j \right]$ | The period when customer demand point $j$ receives service. |
| $c_{p_1}$ | Unit waiting cost for vehicles older than $ET_j$. |
| $c_{p_2}$ | Unit stock-out cost for vehicles later than $LT_j$. |
| $t_{ik}$ | The moment when the kth delivery vehicle departs from node $i$. |
| $x_{ij}^k = \begin{cases} 1 \\ 0 \end{cases}$ | The vehicle travels from node $i$ to node $j$, if its value is 1, otherwise, it is 0. |
| $y_i = \begin{cases} 1 \\ 0 \end{cases}$ | Indicates that the demand of customer $i$ is 1, if it is split, and it is 0. if it is not split. $i \in N$. |
| $y_{ij}^k$ | The load when the kth car is driving on the road $(i, j)$. |
| $z_{ijt}^k = \begin{cases} 1 \\ 0 \end{cases}$ | Decision variable, indicating that the value is 1 when the road $(i, j)$ has vehicle $k$ driving in the period $t$, otherwise, it is 0. |
| $v_{ij}$ | Average vehicle speed. |
| $t_{ijk}$ | Time spent by vehicle $k$ on road $(i, j)$. |
| $\alpha_1$ | Refrigeration cost of a refrigerated truck per unit time during transportation. |
| $C_s$ | The cost of split compensation for splitting the customer's needs once. |

## 3. Construction of Optimization Model for Mixed Transportation Scheme

### 3.1. Objective Function

Equation (1) is the objective function of the mixed transportation scheme optimization model:

$$
\begin{aligned}
minZ &= C_k + C_r + C_f + C_c + C_p + C_{sc} \\
&= \sum_{k \in K} f_k + \sum_{i \in M \cup N} \sum_{j \in M \cup N} \sum_{k \in K} \alpha_1 t_{ijk} x_{ij}^k + \sum_{j \in M \cup N} \sum_{k \in K} \alpha_2 t_j \\
&\quad + \sum_{i \in M \cup N} \sum_{j \in M \cup N} \sum_{k \in K} \sum_{t \in T} v_{ij} t_{ijk} c_t \lambda_c z_{ijt}^k \\
&\quad + \sum_{i \in M \cup N} \sum_{j \in M \cup N} \sum_{k \in K} \sum_{t \in T} v_{ij} t_{ijk} f_t \lambda_f z_{ijt}^k + c_{p_1} \sum_{i=1}^{N \cup S} \max\left\{ ET_i - t_i^k, 0 \right\} + \\
&\quad c_{p_2} \sum_{i=1}^{N \cup S} \max\left\{ t_i^k - LT_i, 0 \right\} + \sum_{j \in N} C_s y_j \left( \sum_{k \in K} \sum_{j \in N} x_{ij}^k - \left\lceil \frac{p_j}{Q} \right\rceil - \left\lceil \frac{q_j}{Q} \right\rceil \right)
\end{aligned}
\tag{1}
$$

Vehicle fixed cost: Vehicle fixed cost refers to the fixed cost incurred by the cold chain transportation enterprise for providing transportation services with refrigerated vehicles, including depreciation costs, driver salaries, rental fees, etc. It is usually constant and independent of the distance traveled by the refrigerated vehicles and the number of customers served. Therefore, the total fixed cost of the vehicle is:

$$C_k = \sum_{k \in K} f_k \tag{2}$$

Refrigeration cost: The refrigeration cost mainly includes two aspects. The first is the cost of refrigeration in the truck compartment. In the transportation process, there is a temperature difference between the outside and inside of the truck compartment. To maintain the temperature inside the compartment, refrigeration is necessary. The second aspect is the additional refrigeration cost incurred when opening the truck door for loading and unloading at customer demand points. Therefore, the total refrigeration cost is the sum of these two parts:

$$C_r = C_{r_1} + C_{r_2} = \sum_{i \in M \cup N} \sum_{j \in M \cup N} \sum_{k \in K} \alpha_1 t_{ijk} x_{ij}^k + \sum_{j \in M \cup N} \sum_{k \in K} \alpha_2 t_j \tag{3}$$

Green cost: Green cost includes two parts, namely vehicle fuel consumption cost and carbon emission cost:
Vehicle fuel consumption cost:

$$C_f = \sum_{i \in M \cup N} \sum_{j \in M \cup N} \sum_{k \in K} \sum_{t \in T} v_{ij} t_{ijk} f_t \lambda_f z_{ijt}^k \tag{4}$$

Carbon emissions cost:

$$C_c = \sum_{i \in M \cup N} \sum_{j \in M \cup N} \sum_{k \in K} \sum_{t \in T} v_{ij} t_{ijk} c_t \lambda_c z_{ijt}^k \tag{5}$$

The time penalty cost caused by the discrepancy between the delivery time and the transportation time window:

$$C_p = c_{p_1} \sum_{i=1}^{N \cup S} max\left\{ ET_i - t_i^k, 0 \right\} + c_{p_2} \sum_{i=1}^{N \cup S} max\left\{ t_i^k - LT_i, 0 \right\} \tag{6}$$

Split compensation costs. When the customer's demand is not split, the split compensation cost due to the split is zero; when the customer's demand is split into $a_s$, the additional cost of receiving, loading, and unloading due to the split demand is called the split cost, which is $C_s \times a_s$. The total number of services for customer $j$ minus the number of services, when the order is not split, is $a_s$. The total split cost increased due to the split is thus:

$$C_{sc} = \sum_{j \in N} C_s y_j \left( \sum_{k \in K} \sum_{j \in N} x_{ij}^k - \left\lceil \frac{p_j}{Q} \right\rceil - \left\lceil \frac{q_j}{Q} \right\rceil \right) \tag{7}$$

*3.2. Constraints*

Equations (8)–(17) are the constraints of the model:

$$\sum_{i \in M \cup N} \sum_{j \in M \cup N} x_{ij}^k y_{ij}^k \leq Q, \forall k \in K \tag{8}$$

$$\sum_{i \in M \cup N} \sum_{k \in K} x_{ij}^k = 1, \forall j \in N \tag{9}$$

$$\sum_{j\in M\cup N}\sum_{k\in K} x_{ij}^k = 1, \forall i \in N \tag{10}$$

$$\sum_{i\in M}\sum_{j\in M} x_{ij}^k = 0, \forall k \in K \tag{11}$$

$$\sum_{i\in M}\sum_{j\in N} x_{ij}^k = \sum_{i\in N}\sum_{j\in M} x_{ij}^k, \forall k \in K \tag{12}$$

$$\sum_{i\in U}\sum_{j\in U} x_{ij}^k \leq |U| - 1, \forall U \in M \cup N, |U| \geq 2, \forall k \in K \tag{13}$$

$$x_{ij}^k\left(t_j^k - t_i^k\right) \geq 0, \forall i, j \in M \cup N, \forall k \in K \tag{14}$$

$$t_j^k = \begin{cases} \left(max\left\{t_i^k, ET_i\right\} + t_i + \frac{D_{ij}}{v_{ij}}\right)x_{ij}^k, & \forall i \in N, \forall j \in M \cup N, \forall k \in K \\ \left(t_{ik} + \frac{D_{ij}}{v_{ij}}\right)x_{ij}^k, & \forall i \in M, \forall j \in M \cup N, \forall k \in K \end{cases} \tag{15}$$

$$y_{ij}^k \geq 0, \forall i, j \in M \cup N, \forall k \in K \tag{16}$$

$$p_i \geq 0, q_i \geq 0, Q \geq 0, \forall i \in M \cup N \tag{17}$$

Equation (8) is the weight limit of refrigerated trucks, which means that the vehicle load of refrigerated truck *k* during transportation cannot exceed the maximum load capacity of refrigerated trucks. Equation (9) indicates that each demand point is only visited once. Equation (10) means that the delivery vehicle must leave after completing the service of the customer node. Equation (11) indicates that there is no situation where the delivery vehicle starts from the distribution center and goes directly to the distribution center. Equation (12) indicates that the delivery vehicle can start from any distribution center. After the delivery task, the driver can return to any distribution center. Equation (13) eliminates the constraints of the branch road. Equation (14) means to strictly follow the driving sequence. Equation (15) refers to the moment when the delivery vehicle k arrives at the demand point j, and $t_j^k$ means to take the maximum value. Equation (16) indicates that the load of the kth delivery vehicle from node *i* to *j* is a non-negative value. Equation (17) indicates that the pick-up and delivery volume at any customer node is a non-negative value, and the delivery maximum load capacity of the vehicle is a non-negative value.

*3.3. Target Solution Form*

$V$: vehicle, $v$: license plate number, $T$: task list, $J$: task list details.
Solution form: $\{V_1, V_2, \ldots, V_n\}$, $V_i = \{v_i, T_i\}$, $T_i = \{J_1, J_2, \ldots, J_m\}$,
$J_i$= (job sequence number, job type, job time, job content, goods/quantity, job location ... ).
Operation types include $M$ (vehicle move), $L$ (load), and $U$(unload).

## 4. Hybrid Heuristic Transportation Scheme Optimization Algorithm

The extended problem of the VRP has been proven to be an NP-hard problem. This study has made many extensions based on this, considering factors such as demand splitting, pickup and delivery, time windows, multiple centers, and semi-openness, which greatly increase the complexity of the model and algorithm. Simply using heuristic algorithms cannot meet the solution requirements and cannot obtain a more valuable set of solutions. Therefore, this paper designs a two-stage hybrid heuristic path solution algorithm that combines tabu search and traversal and genetic algorithms.

### 4.1. Region Segmentation

"Region" refers to the division of geographical space into regions with approximate boundaries according to certain methods and indicators. The characteristics of the region are as follows: the region has a certain boundary; the region has obvious continuity and similarity, and the interconnection between the regions has significant differences. Due to the different purposes, the methods and indicators used are also different, so the areas divided are also different.

#### 4.1.1. Purpose of Regional Segmentation

The concept of "region" is innovatively added to shrink the solution space. Since this article involves the three roles of manufacturer, seller, and distribution center, the needs of customers are complex and numerous, and the locations are widely distributed. If each customer is regarded as an independent individual with regard to the formulation of a transportation plan, the number of solutions cannot be measured, which greatly increases the algorithm time complexity and space complexity. This paper considers that there will be a problem in that the distance between customers in the same area is relatively close in the actual transportation process. If these customers are planned separately, a "garbage solution plan" that "detours" will be formed due to factors such as time windows, as shown in Figure 1. Therefore, this article first divides customers by region, virtualizes a regional center, plans according to the regional center, and then formulates transportation plans for customers within the region according to the time window demands, as shown in Figure 2.

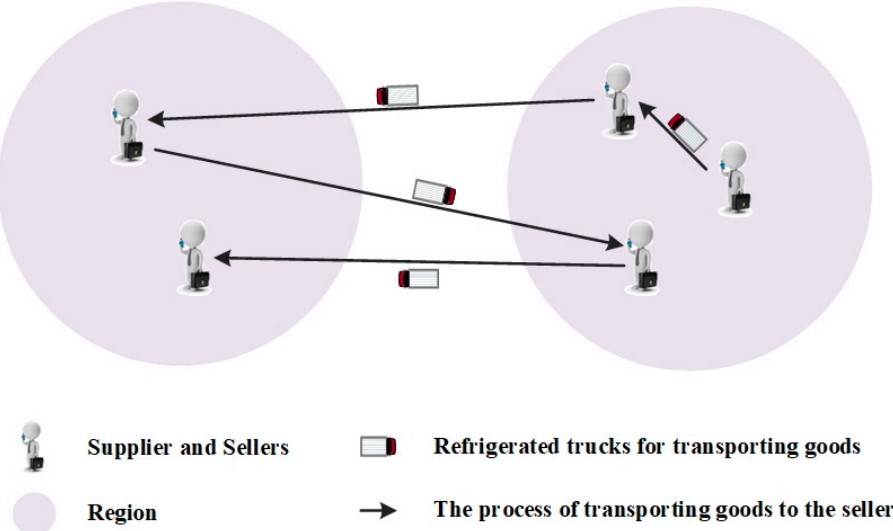

**Figure 1.** Garbage solution plan, garbage solutions due to factors such as time windows.

According to the supplier and sales distribution shown in Figure 2, the advantage of dividing the area is evident. There are 11 customers in total, and they are divided into four regions. We assume that each customer's demand can be split up to a maximum of three times.

Before the area is divided, the time complexity of route planning would be 11! if the customer demand is not split. If each customer demand is split once, the time complexity would be 22!.

After the area is divided, with the route planning method described above, the time complexity for four regions is 4!. Even if a complex traversal search is performed on all customers within each region, the time complexity is 4! + 2! + 3! + 3! + 3!. If each customer demand is split once, the time complexity would be 4! + 4! + 6! + 6! + 6!. Currently, there are only 11 customers, and the time complexity of unsplitted demand is significantly higher than the time complexity of 4! + 2! + 3! + 3! + 3!.

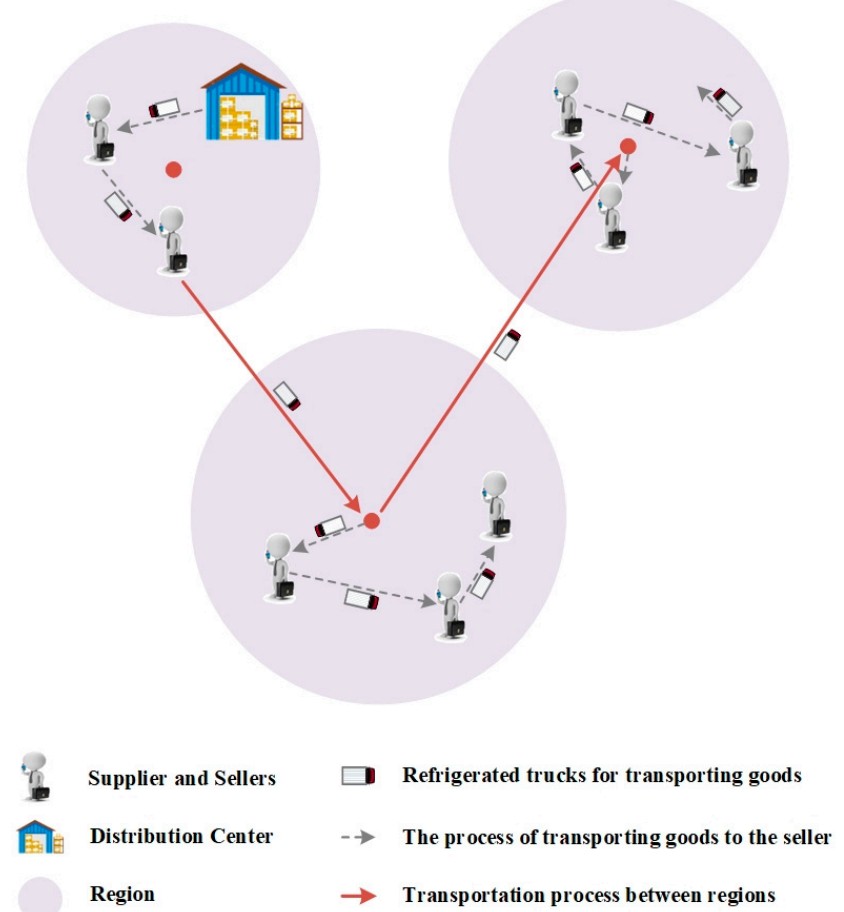

| | | | |
|---|---|---|---|
| 🧑 | Supplier and Sellers | 🔲 | Refrigerated trucks for transporting goods |
| 🏢 | Distribution Center | --→ | The process of transporting goods to the seller |
| 🟣 | Region | ⟶ | Transportation process between regions |

**Figure 2.** Transportation method and solution after introducing regions.

In actual cold chain transport operations, the number of customers is much larger than 11, so to meet the requirements of transportation efficiency, the algorithms should be continuously optimized to reduce the execution time. Therefore, this paper adopts the concept of "region" to greatly reduce the time complexity of cargo loading and route optimization, and to design an optimization algorithm that is suitable for the actual operation of cold chain transportation enterprises.

4.1.2. Region Segmentation Rules

(1) Assumptions: Based on the road conditions of the city, the average driving speed of vehicles is assumed to be 30 km/h. According to the convention, the acceptable delay time range for customers' tasks is within 10 min.

(2) Basic region division: This is divided according to the actual regions of the city (for example, Shanghai is divided into the Xuhui District, Yangpu District, Jing'an District, Minhang District, etc.), which also facilitates the setting of accessible areas for vehicles and solves problems such as urban area restrictions.

(3) Region boundary adjustment: To control the vehicle's travel time between two customers within the same region to no more than 10 min, according to $30 \times 10/60 = 5$, the maximum distance that a vehicle can travel within 10 min is 5 km. Therefore, if the distance between customers within a region is greater than 5 km, the region is divided again.

(4) Dynamic division method: The two customers farthest apart within the region that needs to be redivided are selected as the cluster center points. The k-means algorithm is used to assign all customer nodes within the region to the nearest cluster center point, and the region is eventually divided into two regions.

The flow chart of region segmentation, as shown in Figure 3, assumes that area a contains suppliers $N_1, N_2, N_3, N_4$ and sellers $S_5, S_6, S_7, S_8$, and its spacing set $A$ includes $d_{12}, d_{13}, d_{14}, d_{15}, \ldots, d_{75}, d_{76}, d_{77}$, a total of 56 kinds, known as set $A[56]$.

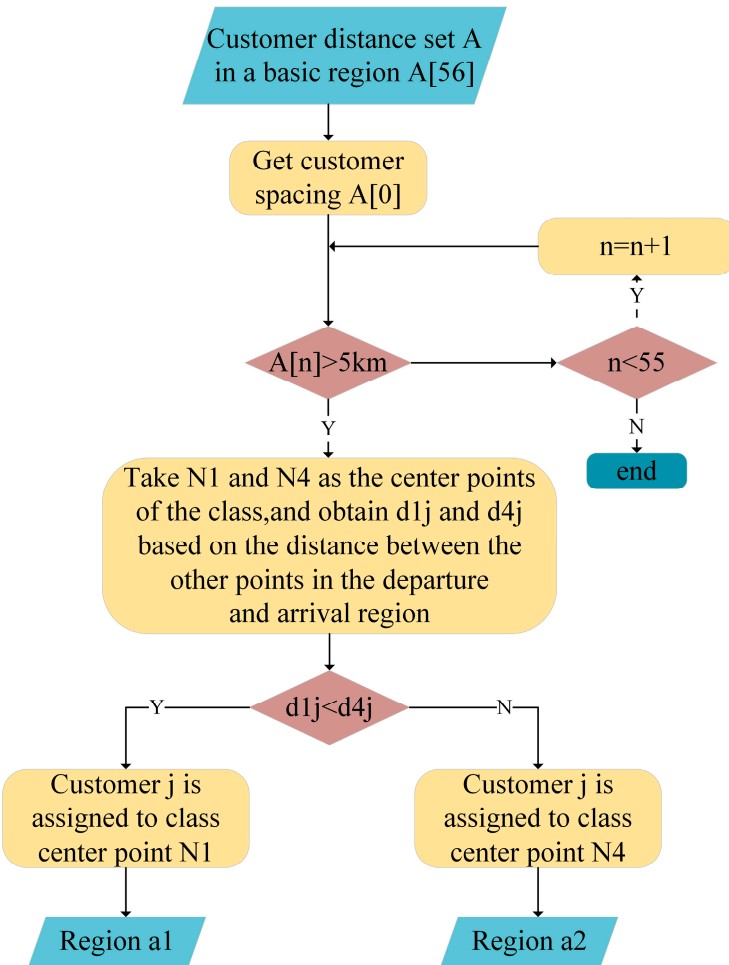

**Figure 3.** Flow chart of regional division indicating the specific region division process.

### 4.2. Customer Demand Partition Mark

$V$: vehicle, $v$: license plate number, $T$: task list, $J$: task list details.
Solution form: $\{V_1, V_2, \ldots, V_n\}$, $V_i = \{v_i, T_i\}$, $T_i = \{J_1, J_2, \ldots, J_m\}$,
$J_i = $ (job sequence number, job type, job time, job content, goods/quantity, job location ... ).
Operation types include $M$ (vehicle move), $L$ (load), and $U$ (unload).

(1)  Customer demand representation: customer demand information includes customer name, origin, origin time window, destination, destination time window, cargo name, cargo weight, whether it can be split, and other information. The following use tasks represent customer demand.

(2)  Relationship between the refrigerated truck lines and customer demand.

Collection of lines: $S = \{L_1, L_2, \cdots, L_i, \cdots, L_k\}$.
Customer demand set for line $i$: $L_i = (task_1, task_2, task_3, \cdots, task_n)$.
Different refrigerated trucks have different driving routes, and different driving routes need to complete different customer demands (tasks). After dividing the region, each customer demand task (origin-destination) is marked as a task (origin region-destination region). After analysis, it was found that many different customer demand tasks have the same mark after marking, and to reduce the search space of the solution, the customer demand tasks are grouped into jobs according to the same driving direction (origin region-destination region). The customer demand task set of a route can then be reduced to a job

sequence; and the job sequence arranged in a certain order represents the regional driving order of the current vehicle. The grouping method is shown in Figure 4.

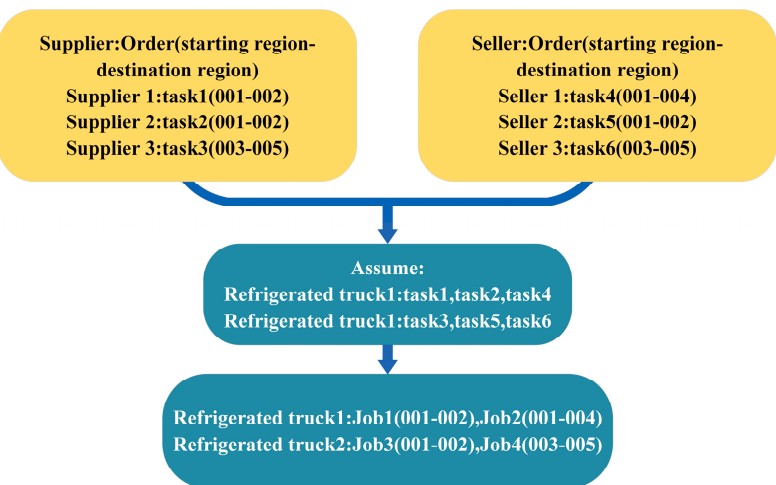

**Figure 4.** Task grouping marked as job list, some tasks are marked with the same characteristics as a job.

*4.3. Two-Stage Hybrid Heuristic Path Optimization Algorithm*

4.3.1. Algorithm Implementation Steps

The core of the solution algorithm designed in this paper is the two-stage genetic algorithm, and the optimal path generation algorithm, the time window constraint algorithm, and the load constraint algorithm are also designed to combine with it.

(1) Since the customer order demand task contains a lot of valid information, it is impossible to obtain a valid solution by using only one customer serial number, so the first stage of the genetic algorithm is a genetic mutation process based on the initial order demand task of the supplier and the seller. Firstly, according to the customer order demand task and the basic parameter information of the reefer truck, the initial solution of the population is generated by using the vehicle proximity principle and the maximum loading principle (the greedy method), and each reefer truck is matched with an unordered set of customer order demand tasks. Since the same region contains multiple suppliers and sellers, each supplier and seller may have multiple order demand tasks, and the order demand may be split; if the customer order demand task is used as the initial data to develop the loading scheme and route optimization, the scale of change of the feasible solution of the algorithm is huge, and the optimal solution cannot be obtained quickly in a predictable time frame. Therefore, the first stage of the genetic algorithm only performs the overall variation among the sets of unordered customer order task sets of multiple reefer trucks and does not involve the ranking order of tasks or the reefer truck driving routes of each reefer truck.

(2) The task groups are marked as job sequences. According to the customer demand task marked after the introduction of the concept of "area" above, it was found that there are many customer orders that require tasks with the same starting area and destination area, so the tasks with the same vehicle driving direction are grouped into a group and expressed as a job sequence. The grouping method is shown in Figure 3.

(3) The second stage of the genetic algorithm is the genetic variation process of the job sequence based on the offspring individuals of the first stage. The order of the job sequence indicates the order of the reefer executing the customer demand and also determines the regional driving order of the reefer, so this paper adopts a special chromosome coding method and genetic variation process and designs a special decoding method: the optimal path generation algorithm, which transforms job sequences into reefer truck driving paths, and at the same time, the optimal path of each job sequence is used as the fitness value for evaluating the merits of the offspring.

(4) Optimal path generation algorithm. The algorithm is designed by a unique method according to the job sequence of the customer demand job sequence, so that the generated vehicle driving paths are all feasible solutions, and the shortest path is found in the feasible solution as the fitness value of the current offspring chromosome.

(5) The load constraint algorithm is combined with the time window constraint algorithm to generate the load scheme. According to the optimal individual (i.e., the job sequence) of the second stage of the genetic algorithm, the optimal path obtained by calling the optimal path generation algorithm is checked by the load constraint algorithm to see if the load is less than the maximum load of the vehicle throughout the whole process, and detailed data such as the load information of each road section of the vehicle, and the loading and unloading information of the customer point are obtained in the process of checking. The time window constraint algorithm designed in this paper is then used to check whether the load scheme meets the time window required by the customer, and the detailed data such as vehicle departure time, driving time, and customer service time are obtained during the checking process; if both are verified to pass, it is been proven that the children obtained in the first stage of the genetic algorithm are feasible solutions, and the detailed vehicle loading scheme is obtained, and the next iteration is carried out; if one is verified to fail, it means that the customer order demand insertion fails, and reselect a new path and repeat the above operation.

### 4.3.2. The First Stage of the Genetic Algorithm

(1) Coding design. The first stage of the genetic algorithm is based on the cross-mutation process of tasks. Due to the diversity of tasks, natural integer encoding is used in this stage, and a chromosome is expressed as (1, 2, 3) and the corresponding matching (*task*1, *task*2, *task*3···). A chromosome corresponds to a reefer truck, and the genes on the chromosome correspond to the set of order requirements that should be fulfilled by that reefer truck.

(2) Population initialization: Generate a feasible initial solution based on the greedy algorithm and Pan Lijun's proposed "time difference method" [41] as the initial population, which can improve the starting point of the genetic algorithm, reduce the number of iterations, and improve the algorithm's performance.

(3) Fitness function: Use the objective function of the "three-in-one" mixed transportation optimization model as the fitness function.

(4) Selection operator: Use the roulette selection method to select individuals, but to avoid losing the optimal individual, introduce the elite strategy (with a parameter of 3) and directly copy the three parent individuals with the smallest fitness function value to the next generation.

(5) Crossover operator: All individuals in the population are randomly paired and new individuals are generated using the single-point crossover principle, as shown in Figure 5.

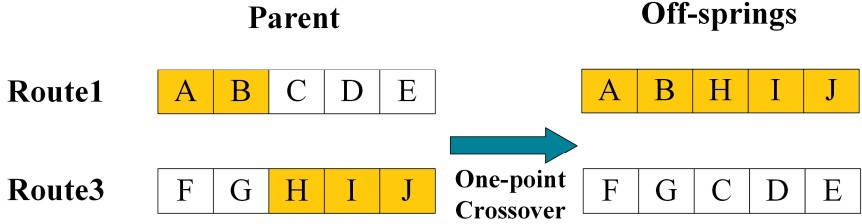

**Figure 5.** The first stage single-point crossover operation. "Route 1" and "Route 3" indicate two paths, and "One-point Crossover" indicates a single-point crossover operation.

(6) Mutation operator. Due to the uniqueness of the initial data, the chromosome coding method, and the solution scheme in this paper, it is very easy to generate invalid offspring by using the conventional mutation operation. To avoid the formation of

infeasible codes, we adopted a simple and efficient mutation operation, as shown in Figure 6.

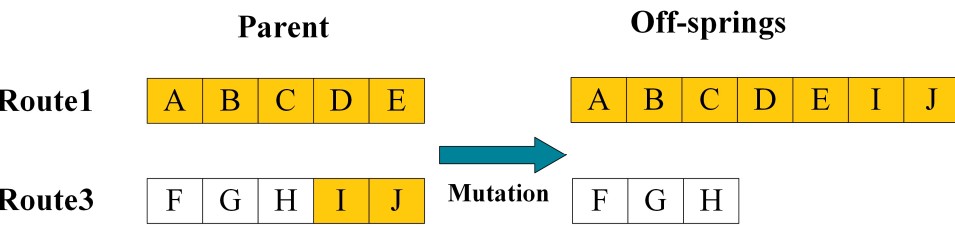

**Figure 6.** The first stage mutation operation. "Route 1" and "Route 3" denote two paths, "Parent" denotes the parent, "Off-springs" denotes the children, and Mutation denotes the mutation operation performed.

First, select a random chromosome (a refrigerated vehicle), select a random gene (a customer order demand task) on that chromosome, and delete it from that chromosome; then select a random one from other chromosomes (other vehicles), and insert the deleted gene (customer order demand task) into the current chromosome. First, we should judge that the cargo class of the genetic gene to be inserted is the same as the cargo class of the chromosome, and then continue the subsequent operation; otherwise, the insertion fails.

(7)  Termination condition one is to set the maximum number of iterations of the first stage genetic algorithm to 150, and termination condition two is that the optimal solution still does not produce any changes after 10 consecutive iterations; if the termination condition is satisfied, the algorithm optimization process is terminated, and the optimal cargo solution and vehicle driving route are output.

### 4.3.3. Task Composition Transformation Job Sequence

As revealed in the first stage of the genetic algorithm, each chromosome in the current population contains multiple genetic genes, representing the fulfillment of multiple customer demands (tasks) by a refrigerated truck. To simplify the chromosome representation, customer demand tasks are categorized by their region and direction (start region-end region) into a group called a job. Thus, a set of customer demand tasks on a single chromosome can be condensed into a job sequence. This job sequence serves as the input data for the second stage of the genetic algorithm. The transformation process is illustrated in Figure 4.

### 4.3.4. The Second Stage of the Genetic Algorithm

Transform the tasks of the first stage genetic algorithm into jobs and remove the duplicates in order to reduce the search space and improve the efficiency of the algorithm. The converted data will serve as the input for the second stage genetic algorithm.

(1)  Coding design. The second stage genetic algorithm is based on job sorting for genetic variation. In encoding, the path problem studied in this paper is influenced by the transportation requirements, where each transport task has a fixed origin and destination, and the transportation process has strong directionality. Binary and natural integer encoding can easily produce invalid solutions. Therefore, a character sequence-based encoding method is used, where jobs are used as genetic units and a chromosome can be represented as (001-002; 003-004 . . . ). This encoding method is beneficial for generating paths.

(2)  Fitness function. We used the optimal path generation algorithm. Each job sequence can generate multiple vehicle travel routes, and the job sequence is evaluated according to the length of the optimal route.

(3)  Select operator. Select the job sequence with the smallest fitness function value in the candidate set.

(4)    Mutation operator. The order of jobs in a job sequence determines the order in which vehicles perform tasks; that is, the regional driving path. Therefore, the second stage of the genetic algorithm is to mutate the job order on a chromosome to generate new individuals. The mutation process is as follows: randomly select two jobs on the same chromosome to exchange, as shown in Figure 7.

**Parent**                                                                    **Off-springs**

Route1 | Job1 | Job2 | Job3 | Job4 | Job5 | → **Mutation** | Job5 | Job2 | Job3 | Job4 | Job1

**Figure 7.** The second stage mutation operation. "Route 1" denotes the path, "Parent" denotes the parent, "Off-springs" denotes the children, and Mutation denotes the mutation operation performed.

(5)    Termination conditions. If the optimal chromosome of the current population still does not produce changes in six consecutive iterations, the second stage of the genetic algorithm is terminated and the loading scheme is generated.

### 4.3.5. Optimal Path Generation Algorithm

The optimal path generation algorithm (*best_route*) is a method for calculating the fitness value of the second-stage genetic algorithm, and it is also the core algorithm for solving the cold chain logistics load scheme and route optimization model. The algorithm parameters are shown in Table 3. The basic logic of *best_route* is shown in Algorithm 1.

---

**Algorithm 1:** Optimal Path Generation Algorithm (*best_route*)

---

**Input:** *turns*, *sr*, *regions*, $L_i$, $L_jij$
**Output:** The sequence of vehicle driving paths in the effective region.
1: **if** $i < L_i$, **then**
2:    $C_i = L_i + 1$
3:    **while** $C_i < regions + 1$ **do**
4:       Insert the new Job into the path sequence *sr*
3:       **if** $j \geq C_i$, **then**
4:          Output the sequence of vehicle driving paths in the current region
5:       **else**
6:          $C_j = C_i + 1$, The destination region of the new Job is one digit after the start region.
7:          **while** $C_j \leqq regions + 1$ **do**
8:          Insert the new Job into the path sequence *sr*, output the sequence of vehicle driving paths in the current region
9:             $C_j = C_j + 1$
10:          **end while**
11:          $C_i = C_i + 1$
12:    **end while**
13: **else**
14:    $C_i = i$
15::    **if** $j > C_i$    **then**
16:       Output the sequence of vehicle driving paths in the current region
17::    **else**
18:       $C_j = C_i + 1$
19:       **while** $C_j < regions + 1$ **do**
20:          Insert the new Job into the path sequence *sr*, output the sequence of vehicle driving paths in the current region
21:          $C_j = C_j + 1$
22:       **end while**
23: **end if**

---

**Table 3.** Algorithm parameters (*best_route*).

| Parameter | Description |
|---|---|
| S_JOBS | The current complete *Job* sequence. |
| turns | The *Job* number of the current path to be inserted in the *Job* sequence. |
| sr | The optimal path after the las *Job* insertion is completed. |
| Li | The position of the starting region of the previous *Job* in the current optimal path *sr*, indicated by a numeric sequence. |
| Lj | The position of the destination region of the previous *Job* in the current optimal path *sr*, indicated by a numeric sequence. |
| i | The position in *sr* of the starting region of the *Job* whose path is to be inserted (before the start of the algorithm), indicated by a numerical number. If there is no starting region of the *Job* in *sr*, *i* is 0. |
| j | The position in *sr* of the destination region of the current *Job* to be inserted into the path (before the start of the algorithm), expressed as a numerical number. If there is no destination region of the *Job* in *sr*, *j* is 0. |
| regions | Total number of regions of the current optimal path *sr*. |
| Ci | The location of the start region of the *Job* where the current path will be inserted (during the algorithm). |
| Cj | The location of the destination region of the current *Job* that will be inserted into the path (during the algorithm). |

### 4.3.6. Load Capacity Constraint Algorithm

After the genetic mutation process of the two-stage genetic algorithm, we verify whether the offspring of the second-stage genetic algorithm is a feasible solution. One of the constraints is to judge whether each refrigerated truck is overloaded during transportation, but the cold chain logistics process designed in this paper happens between the three roles of the supplier, distribution center, and seller, which means that there are both unloading and loading services during the transportation of refrigerated trucks, so the total load of the vehicle may exceed the maximum load.

In view of the complex transportation process in this paper, we introduce the concept of "road section" in the load constraint algorithm. For example, if the vehicle travel path is 001-003-002, it is first decomposed into road sections 001-002 and 002-003, and then the refrigerated truck travels from area 001. Whether there is overloading when traveling to area 003, and whether there is overloading when driving from area 003 to area 002, if each road section meets the load constraint, the whole vehicle transportation process is reasonable and feasible. Apply the above operations to the load constraint algorithm, the specific operations of which are as follows:

(1) According to the offspring *Job* sequence with the smallest fitness value in the candidate set of the genetic algorithm in the second stage, call the optimal path generation algorithm (*best_route*) to obtain the optimal route.
(2) Decompose the optimal path into an array of "segments".
(3) Construct a road section cargo loading plan. Proceed as follows:

Step 1: Traverse the "road section" array, and select each road section in turn;
Step 2: Match the start region, destination region, and road segment start region and destination region of each job in the *Job* sequence required by the customer order. If the *Job* start region is the same as the road segment start region, insert this *Job* into the road segment. The vehicle's current add the cargo weight of this *Job* to the load capacity. If the destination region of the *Job* is the same as the starting area of the road segment, delete this *Job* from the customer order demand of this road segment.
Step 3: According to the increase or decrease of the vehicle load in Step 2, the vehicle load of each section is calculated and compared with the maximum load of the vehicle.

An example to illustrate the execution process of the load constraint algorithm: Assume that the maximum load capacity of the vehicle is 3 t, the order requirement of sellers includes $Job_1$ = (region 001-region 003, cargo weight 1 t), and $Job_2$ = (region 001-region 005, cargo weight 1 t). The order requirement of suppliers is $Job_3$ = (region 003-region 002, cargo weight 3 t), and the optimal path is 001-003-002-005. Therefore, the road section loading

scheme is constructed, as shown in Table 4, in which L means Load loading, and U means Unload unloading.

**Table 4.** Loading plan of the road section.

| Road Section | Customer Order Requirements | Vehicle Load (t) |
|---|---|---|
| 001-003 | $Job_1$ (L), $Job_2$ (L) | $1 + 1 = 2$ |
| 003-002 | $Job_1$ (U), $Job_3$ (L) | $2 - 1 + 3 = 4$ |
| 002-005 | $Job_1$ (U) | $4 - 3 = 1$ |

It can be seen from the road section cargo loading scheme that the vehicle load in section "003-002" is greater than the maximum vehicle load, and overload occurs. Therefore, the current cargo loading scheme does not meet the vehicle load constraints, which is unreasonable. Faced with such a situation, there are two ways to deal with it. The first is that if the customer's order requirement, $Job_1$ does not allow the splitting of the requirement, then this solution is unfeasible; the second way is that if the customer's order requirement $Job_3$ allows for the splitting of the requirement, then split $Job_3$ into two parts; one part is the maximum load capacity of the vehicle minus the used load capacity, and the other part is $3 - 2 = 1$ t, and this part is used as the new supplier order demand $Job_4$ (region 003-region 002, cargo Weight 1 t) is inserted into the loading scheme of other vehicles. The loading scheme of the road section after the current vehicle is updated is shown in Table 5.

**Table 5.** Updated plan of the road section.

| Road Section | Customer Order Requirements | Vehicle Load (t) |
|---|---|---|
| 001-003 | $Job_1$(L), $Job_2$(L) | $1 + 1 = 2$ |
| 003-002 | $Job_1$(U), $Job_3$(L) | $2 - 1 + 2 = 3$ |
| 002-005 | $Job_1$(U) | $3 - 3 = 0$ |

In summary, the load constraint algorithm designed in this paper can not only judge whether the offspring of the two-stage genetic algorithm is a feasible solution, but can also provide a direct basis for the demand splitting of the load scheme. The detailed vehicle pre-allocation plan is shown in Table 6, and M in the table represents the moving process of the vehicle.

**Table 6.** Vehicle pre-allocation plan.

| Vehicle Code | Service Order Number | Customer Order Requirements | Service Type | Starting Location | Destination | Order Quantity | Vehicle Load (t) |
|---|---|---|---|---|---|---|---|
| 1 | 10 | $Job_1$ | L | 001 | 003 | 1 t | 1 |
| 1 | 20 | $Job_1$ | L | 001 | 005 | 1 t | $1 + 1 = 2$ |
| 1 | 30 | | M | 001 | 003 | | 2 |
| 1 | 40 | $Job_1$ | U | 001 | 003 | 1 t | $2 - 1 = 1$ |
| 1 | 50 | $Job_1$ | L | 003 | 002 | 2 t | $1 + 2 = 3$ |
| 1 | 60 | | M | 003 | 002 | | 3 |
| 1 | 70 | $Job_1$ | U | 003 | 002 | 2 t | $3 - 2 = 1$ |
| 1 | 80 | | M | 002 | 005 | | 1 |
| 1 | 90 | $Job_1$ | U | 001 | 005 | 1 t | $1 - 1 = 0$ |

### 4.3.7. Time Window Constraint Algorithm

The time window constraint algorithm further processes the result of the load constraint algorithm; the vehicle pre-allocation plan judges whether the current plan meets the customer's time window requirements and generates the final vehicle loading plan. According to the starting region, the destination area, and the operation type of the vehicle, we divide the customer order types into four categories: loading in the same region, unloading in the same region, loading in the same region, and unloading in the same region, using four different time windows constraint algorithms further processes the

vehicle provisioning plan. Algorithms 2 and 3 represent the basic logic implementation of loading and unloading goods in the same region. The algorithm process of cross-region loading is the same as that of same-region loading. The difference is that the calculation method of vehicle arrival time is different. Lines 4, 9, and 18 of Algorithm 2 are changed to $END\_TIME = begin\_t\_time +$ cross-region vehicle travel time. The algorithm process of inter-area unloading is the same as that of unloading in the same region. The difference is that the calculation method of vehicle arrival time is different. Change the fifth line of Algorithm 2 to $END\_TIME = begin\_t\_time +$ cross-region vehicle travel time. The algorithm parameters are shown in Table 7.

---

**Algorithm 2:** Time Window Constraint Algorithm (The loading and unloading plan is "loading in the same region")

---

**1: if** The current task is the first task (task number = 1) **then**
**2:**    **if** Currently in the first segment **then**
**3:**        $BEGIN\_TIME = begin\_t\_time$
**4:**        $END\_TIME = begin\_t\_time + 10$ min
**5:**    **else** Currently not the first segment
**6:**        $BEGIN\_TIME =$ Vehicle end time of the previous section
**7:**        Get the previous segment $LAST\_AREA$
**8:**        Calculate *travel_time* according to the distance between $LAST\_AREA$
            and $BEGIN\_AREA$
**9:**        $END\_TIME= begin\_t\_time+travel\_time$
**10:**    **end if**
**11:**    $TASK\_TYPE = M$
**12:**    Add *tmp_c_transport_task*
**13:**    task number + 1
**14: end if**
**15: if** $LAST\_AREA != BEGIN\_AREA$ **then**
**16:**    $TASK\_TYPE = M$
**17:**    $BEGIN\_TIME =$ Vehicle end time of the previous section
**18:**    $END\_TIME= BEGIN\_TIME + 10$ min
**19:**    Add *tmp_c_transport_task*
**20:** task number + 1
**21: end if**
**22: if** the current time is more than 24 h old **then**
**23:**    return $-1$(The time window does not match, insert failed)
**24: end if**
**25:** $TASK\_TYPE = L$
**26: if** $END\_TIME < early\_time$ **then**
**27:**    Current task loading start time $= early\_time$
**28:**    Current task loading end time $= early\_time + 10$ min
**29:**    Add *tmp_c_transport_task*
**30:**  **else**
**31:**    Current task loading start time $= END\_TIME$
**32:**    Current task loading end time $= END\_TIME + 10$ min
**33:**    Add *tmp_c_transport_task*
**34: end if**

---

**Table 7.** Time window constraint algorithm parameters.

| Parameter | Description | Parameter | Description |
|---|---|---|---|
| *ROUTE_NO* | Route number. | *TASK_TYPE* | Job type. |
| *M* | Vehicle movement. | *L* | Loading. |
| *U* | Unload. | *S_ROUTE* | Route. |
| *PLATE_NO* | License plate number. | *s_l_address* | The current departure place of the vehicle. |
| *begin_t_time* | Vehicle departure time. | *tmp_c_transport_task* | Vehicle loading schedule. |

To sum up, the time window constraint algorithm designed in this paper can not only judge whether the offspring of the two-stage genetic algorithm is a feasible solution, but also generate a complete vehicle loading scheme during the algorithm execution process. This is assuming that $Job_1$ includes $Task_1$ and $Task_2$, $Job_2$ includes $Task_1$, and $Job_3$ includes $Task_1$, $Task_5$, and $Task_6$. The completed vehicle loading scheme is shown in Table 8.

---

**Algorithm 3:** Time Window Constraint Algorithm (The loading and unloading plan is "Unload in the same region")

---

**1:** task number + 1
**2: if** *LAST_AREA*! = *BEGIN_AREA* **then**
**3:**    *TASK_TYPE* = M
**4:**    *BEGIN_TIME* = Vehicle end time of the previous section
**5:**    *END_TIME* = *BEGIN_TIME* + 10 min
**6:**    Add *tmp_c_transport_task*
**7:**    task number + 1
**8: end if**
**9: if** the current time is more than 24 h old **then**
**10:**     return − 1 (The time window does not match, insert failed)
**11: end if**
**12:**    *TASK_TYPE* = U
**13: if** *END_TIME* < *early_time* **then**
**14:**    Current task unload start time = *early_time*
**15:**    Current task unload end time = *early_time* + 10 min
**16:**    Add *tmp_c_transport_task*
**17: else**
**18:**    Current task unload start time = *END_TIME*
**19:**    Current task unload end time = *END_TIME* + 10 min
**20:**    Add *tmp_c_transport_task*
**21: end if**

---

**Table 8.** Vehicle loading scheme.

| Service Order Number | Customer Order Requirements | Service Type | Starting Region | Starting Location | Target Region | Destination | Order Quantity | Vehicle Load (t) |
|---|---|---|---|---|---|---|---|---|
| 10 | $task_1$ | L | 001 | 01 | 003 | 07 | 0.5 t | 0.5 |
| 20 | $task_2$ | L | 001 | 01 | 003 | 08 | 0.5 t | 1 |
| 30 | | M | 001 | 01 | 001 | 02 | | 1 |
| 40 | $task_3$ | L | 001 | 02 | 005 | 09 | 0.5 t | 1 + 1 = 2 |
| 50 | | M | 001 | 02 | 003 | 07 | | 2 |
| 60 | $task_1$ | U | 001 | 01 | 003 | 07 | 0.5 t | 2 − 0.5 = 1.5 |
| 70 | | M | 003 | 07 | 003 | 08 | | 1.5 |
| 80 | $task_2$ | U | 001 | 01 | 003 | 08 | 0.5 t | 1.5 − 0.5 = 1 |
| 90 | | M | 003 | 08 | 003 | 05 | | |
| 100 | $task_4$ | L | 003 | 05 | 002 | 04 | 0.5 t | 1 + 1 = 2 |
| 110 | $task_5$ | L | 003 | 05 | 002 | 04 | 0.5 t | 2 + 0.5 = 2.5 |
| 120 | $task_6$ | L | 003 | 05 | 002 | 03 | 0.5 t | 2.5 + 0.5 = 3 |
| 130 | | M | 003 | 05 | 002 | 03 | | 3 |
| 140 | $task_6$ | U | 003 | 05 | 002 | 03 | 0.5 t | 3 − 0.5 = 2.5 |
| 150 | | M | 002 | 03 | 002 | 04 | | 2.5 |
| 160 | $task_4$ | U | 003 | 05 | 002 | 04 | 0.5 t | 2.5 − 1 = 1.5 |
| 170 | $task_5$ | U | 003 | 05 | 002 | 04 | 0.5 t | 1.5 − 0.5 = 1 |
| 180 | | M | 002 | 04 | 005 | 09 | | 1 |
| 190 | $task_3$ | U | 001 | 02 | 005 | 09 | 0.5 t | 1 − 1 = 0 |

### 4.3.8. Algorithm Performance Improvement

The time complexity and space complexity of an algorithm can reflect the pros and cons of the algorithm to a large extent. Cold chain logistics companies need to carry out daily transportation planning, which has extremely high requirements for algorithm efficiency. In addition to the "customer regionalization" method mentioned above, this paper also adopts the method of "trading space for time" to reduce the time complexity and improve algorithm efficiency.

(1)  Introduce a "taboo list". Use the tabu table to save the recent crossover or mutation operation so that it cannot be used as the next search direction of genetic optimization, and avoid the optimization process from falling into a loop.

(2) Utilize the storage and memory function of the database. In the first stage of the genetic algorithm, compare the task sequences of the current population with the task sequences of individuals in the database table. If there is a matching sequence, assign the fitness value of the individual in the database table to the individual in the current population directly. In the second stage of the genetic algorithm, compare the job sequences of the current population with the job sequences of individuals in the database table. If there is a matching sequence, assign the optimal route and fitness value of the individual in the database table to the individual in the current population. Therefore, this approach takes up more physical space but avoids redundant calculations, saving computation time.

(3) Apply Python to realize parallel computing. Incorporating parallel computing modules and genetic optimization at the same time increases the space complexity of the algorithm, but greatly reduces the time complexity and speeds up the solution.

## 5. Application Case Analysis

In order to test the validity of the model and algorithm, this paper uses data from a cold chain logistics company in Shanghai. The company has two distribution centers (Pudong cold storage and Jing'an cold storage), and provides "pick-up" service for manufacturers and "delivery" service for sellers at the same time, and currently needs to process 32 delivery orders and 18 pick-up orders from 18 customer points (different orders cannot be mixed according to the type of goods). The orders are not mixed according to the different types of goods The orders of each customer point are shown in Table 9 below; the data in the table are excerpted data, and each piece of data contains information about the starting point, end point, customer time window, goods, the number of goods, whether the goods can be mixed, and whether the demand can be split.

**Table 9.** Information about orders from sellers and manufacturers (excerpts).

| Starting Point | End | Client Point Time Window | Goods | Number of Pieces | Demand | Can It Be Mixed | Can It Be Split |
|---|---|---|---|---|---|---|---|
| Pudong Cold Storage | Xinduhui Food Street | 05:30–08:30 | Fish | 15 | 600 | No | Yes |
| Jing'an Cold Storage | Golden Bridge Shopping Center | 17:00–20:00 | Banana | 12 | 600 | No | Yes |
| Xinduhui Food Street | Pudong Cold Storage | 05:30–20:00 | Soy Sauce | 20 | 200 | Yes | Yes |
| Golden Bridge Shopping Center | Jing'an Cold Storage | 05:00–20:00 | Banana | 30 | 300 | No | Yes |
| Metro | Pudong Cold Storage | 06:00–08:00 | Shrimp | 1 | 50 | No | No |
| Hang Lung Plaza | 889 Square | 05:00–20:00 | Pitaya | 20 | 200 | No | Yes |

The grid distribution map and area division of customer points and distribution center geographic locations are shown in Figure 8 below. The rules of area division are shown in Figure 3, and Figure 8 indicates that all customer points are divided into six regions. It is known that the fixed cost $C_k$ of the vehicle is CNY 500, the maximum loading capacity $Q$ is $2t$, the inter-regional distance data is consistent with the actual road distance, the average vehicle speed $v$ is 30 km/h, the unit fuel consumption cost $\lambda_f$ is 2 yuan/L, and the average fuel consumption rate per unit distance $f_t$ is 1.5 L/km, the unit carbon emission cost $\lambda_c$ is CNY 0.1/kg, the carbon emission rate $c_t$ is 3.91 kg/km, the refrigeration cost $a_1$ during the transportation is 2 yuan/h, and the refrigeration cost $a_2$ during loading and unloading is CNY 4/h, the waiting cost of the early unit is CNY 5/h, the out-of-stock cost of the late unit is CNY 10/h, and the compensation cost for one split is CNY 200.

The multi-objective optimization model is solved using the hybrid heuristic transportation scheme solution algorithm studied in this paper to obtain the optimal path and minimum distribution cost for each vehicle, as shown in Table 10 below. The experimental results include the service order of customers as well as the driving path of the reefer trucks. The driving path of one of the reefer trucks is displayed on the map, and the results are shown in Figure 9.

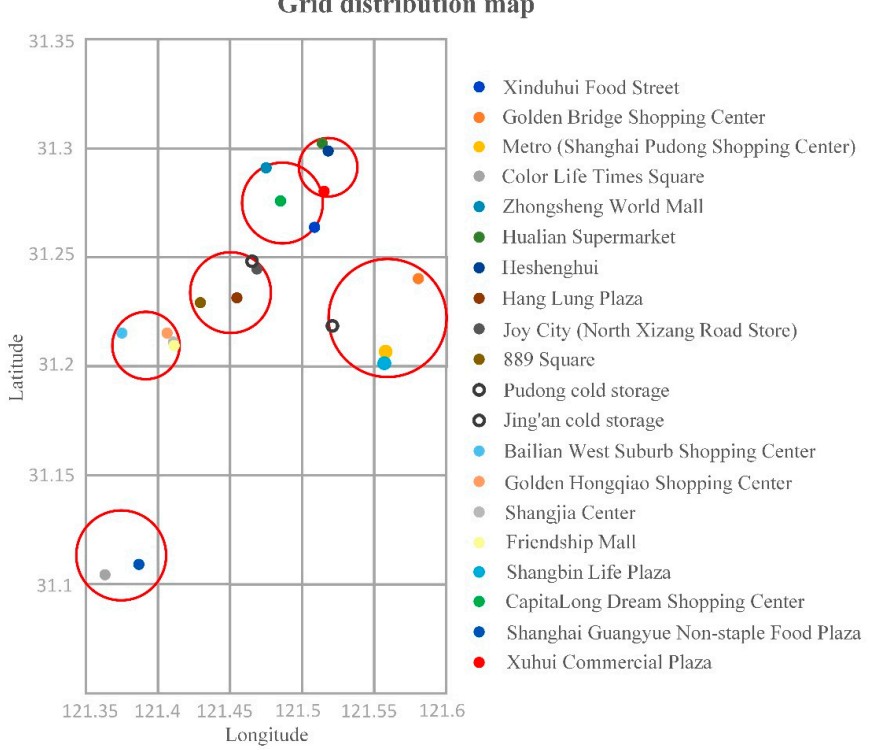

**Figure 8.** Grid distribution of customers and distribution locations.

**Table 10.** Algorithm solution results.

| Vehicle | Customer Service Sequence | Mileage (km) | Vehicle Utilization | Total Delivery Cost (Yuan) |
|---|---|---|---|---|
| 1 | 02-14-15-01-11-02 | 87.238 | | |
| 2 | 02-17-5-01-7-8-01 | 76.194 | | |
| 3 | 02-6-8-13-4-01 | 88.298 | | |
| 4 | 02-14-4-5-12-02 | 81.458 | | |
| 5 | 02-3-01-3-01-9-02-7-18-01-2-01-8-02 | 86.391 | 82.5% | 13,116.776 |
| 6 | 01-1-01-16-15-01-13-02 | 102.431 | | |
| 7 | 01-1-01-5-6-18-02-16-01-9-02 | 91.7 | | |
| 8 | 02-10-01-12-2-17-14-02 | 73.042 | | |

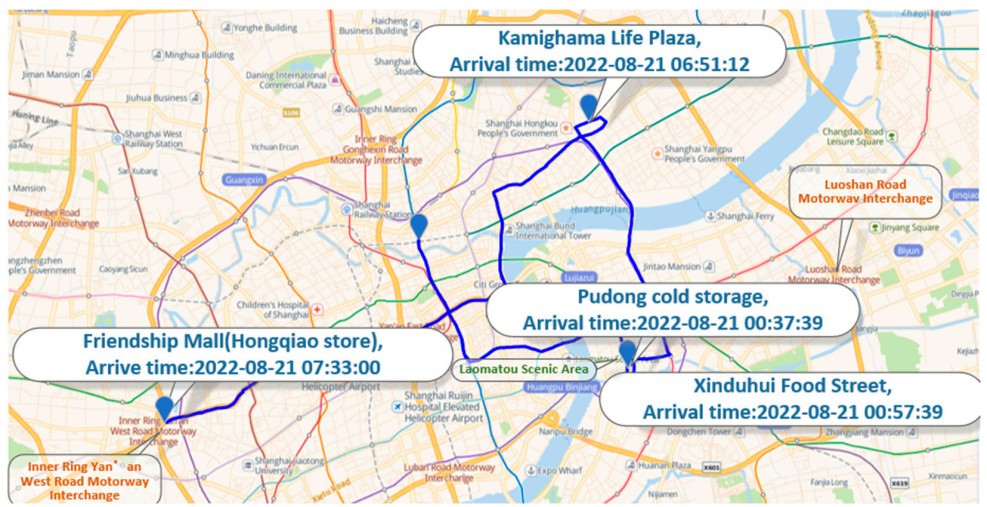

**Figure 9.** Map showing transport routes.

Using the hybrid heuristic transportation scheme-solving algorithm studied in this paper to solve the multi-objective optimization model, the optimal path, minimum cost (Table 10), and loading scheme of the whole transportation process (Table 9) for each vehicle are obtained, and it is displayed on the map (as shown in Figure 9 below). It can be seen that the transportation process after algorithm optimization makes the total distribution cost controlled at about CNY 7000 (including the vehicle fixed cost of CNY 4000, fuel consumption cost of CNY 2060, carbon emission cost of CNY 268, refrigeration cost of CNY 324, and a time penalty cost of CNY 264. The split cost is CNY 200), and the vehicle utilization rate reaches 82.5%. Figure 10 shows the number of iterations of the algorithm in the current case. It can be seen that the optimal solution can be reached about 50 times.

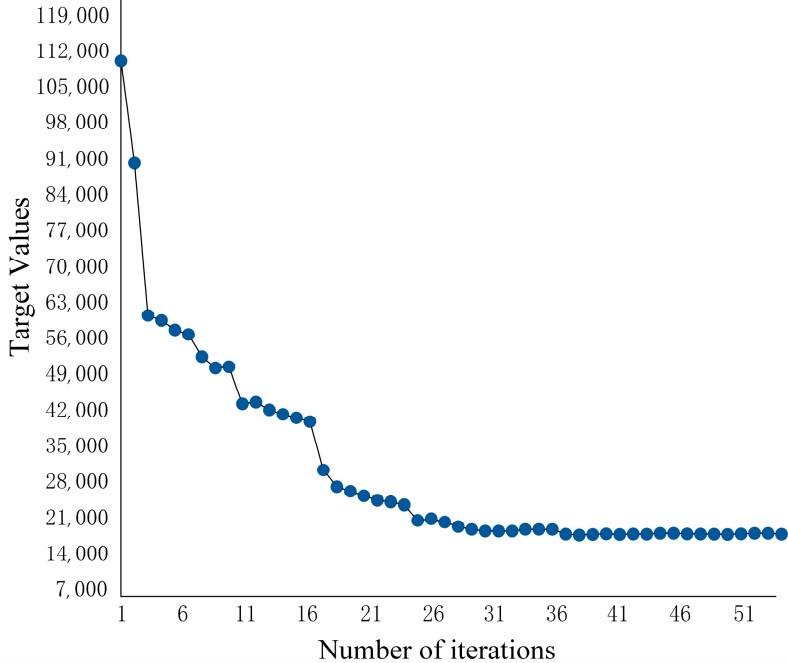

**Figure 10.** Total Cost Convergence.

Table 11 shows the loading scheme of the whole transportation process (excerpted data). The loading scheme includes the license plate number of the refrigerated truck, the wave of the transported cargo, the task type, the task start time, the task end time, the task start point, the task endpoint, and the service volume of each stage. The results can provide cold chain logistics companies with exact transport solutions.

**Table 11.** Loading scheme for the whole transportation process (excerpt).

| Number Plate | Stage | Sequence Code | Task Type | Task Start Time | Task End Time | Starting Point | End | Demand |
|---|---|---|---|---|---|---|---|---|
| 1 | 1 | 20 | L | 21 August 2022 0:35 | 21 August 2022 0:45 | Jing'an Cold Storage | Shangjia Cente | 600 |
| 1 | 1 | 30 | L | 21 August 2022 0:45 | 21 August 2022 0:55 | Jing'an Cold Storage | Friendship Mall | 300 |
| 1 | 1 | 40 | L | 21 August 2022 0:55 | 21 August 2022 1:05 | Jing'an Cold Storage | Friendship Mall | 300 |
| 1 | 1 | 50 | M | 21 August 2022 1:05 | 21 August 2022 1:20 | Jing'an Cold Storage | Shangjia Cente | |
| 1 | 1 | 60 | U | 21 August 2022 8:00 | 21 August 2022 8:10 | Jing'an Cold Storage | Shangjia Cente | 600 |
| 1 | 1 | 70 | M | 21 August 2022 8:10 | 21 August 2022 8:25 | Shangjia Center | Friendship Mall | |
| 1 | 1 | 80 | U | 21 August 2022 8:25 | 21 August 2022 8:35 | Jing'an Cold Storage | Friendship Mall | 300 |

Figure 10 shows the number of iterations of the algorithm for the current case. When we solve the problem using two-stage genetics, the optimal solution is reached in about 50 iterations, which is a significant effect of the algorithm.

In addition, by comparing the time complexity, optimization process, and the results of the algorithm with and without the "region" concept in both parallel and non-parallel computing cases, as shown in Table 12, it is concluded that the objective value obtained without the "region" concept is slightly smaller than that obtained with the "region" concept, but the resulting time cost is unacceptable for the actual operation of the enterprise, so the algorithm with the "region" concept is superior in the case of parallel computing. Therefore, the algorithm with the concept of "region" is more superior in the case of parallel computing, and the superiority of "region" will be more obvious as the volume of data becomes larger.

**Table 12.** Comparison of algorithms in different situations.

| | Not Divided into Regions (No Parallelism) | Divided into Regions (No Parallelism) | Not Divided into Regions (Parallelism) | Divided into Regions (Parallelism) |
|---|---|---|---|---|
| Algorithm Time Complexity | $O(n!)$ | $O(m! + m * \frac{n}{m}!)$ | $O\left(\frac{n!}{2}\right)$ | $O((m! + m * \frac{n}{m}!)/2)$ |
| Algorithm execution time | 123.3 min | 12.3 min | 68.1 min | 5.4 min |
| Target optimal solution | 12,909.5 | 13,220.7 | 12,897 | 13,116.776 |

n: the total number of suppliers and vendors; m: the assumption that all customers are divided into m regions equally; according to the real situation, the aggregation of business districts makes the geographic location of customers generally more dense, so n >> m; currently using 2 CPUs for parallel computation of the model. In the actual operation of the algorithm, because the vehicle travel time of customer spacing in the same area is less than 10 min, the cost gain brought by vehicle path planning for customers in the same area is very small, so it can be ordered according to the customer time window, and the time complexity $O\left(m * \frac{n}{m}!\right)$ is negligible.

As can be seen from Table 10, after the algorithm optimization of the total transportation process, the total distribution cost is about CNY 13,000 (including the vehicle fixed cost of CNY 4000, fuel costs of CNY 2339, the carbon emission cost of CNY 305, the cooling cost of CNY 513, the time penalty cost of CNY 564, and the splitting compensation cost of CNY 5000). The reason for more splitting compensation costs is that the same customer has to split due to the different cargo types of orders submitted. The total vehicle utilization rate reaches 82.5%. The total distribution cost and vehicle utilization rate before and after algorithm optimization are shown in Table 13.

**Table 13.** Comparison of total cost and vehicle utilization before and after optimization.

| | Total Distribution Cost (CNY) | Vehicle Utilization |
|---|---|---|
| Not optimized | 20,000 | 50% |
| After optimization | 13,000 | 82.5% |

Compared with the actual total cost of distribution of nearly CNY 20,000, the solution result saves nearly 35%; moreover, as the number of customer orders increases, the solution advantage of the algorithm designed in this paper will be more obvious.

## 6. Conclusions and Prospects

### 6.1. Conclusions

Based on considering the impact of manufacturers joining the overall logistics distribution, we propose a two-stage partitioning strategy based on multiple distribution centers and demand splitting. The strategy combines vehicle load capacity, mixed cargo restrictions, and service time window constraints, and provides services in multiple waves.

We fully consider factors such as vehicle travel distance, waiting time, and vehicle occupancy rate, and construct a mathematical model with the sum of fixed vehicle cost, green cost, refrigeration cost, time penalty cost, and split compensation cost as the optimization objective. To deal with the complexity of the model and the uniqueness of the solution, we designed a two-stage hybrid heuristic path solution algorithm combining tabu search and a genetic algorithm. We obtained more practical solutions and presented the path planning results as a web map in a more intuitive way. Finally, we conducted a verification analysis of the model's feasibility and the algorithm's effectiveness through an actual case study. The experimental results show that the mixed transportation scheme optimization model proposed in this paper can sufficiently solve the cold chain logistics transportation problem under complex conditions. This model can effectively improve vehicle utilization, reduce driving paths, save transportation costs and inventory costs, and improve customer satisfaction. The two-stage hybrid heuristic algorithm solution algorithm proposed in this paper can better solve the problem model, further improve the solution speed, result in a more reasonable optimal solution, and the algorithm performance has been effectively verified.

*6.2. Prospects*

For the cold chain logistics vehicle loading scheme and route optimization problem under the influence of complex demand, this paper has already achieved certain results. However, due to the special characteristics of cold chain logistics, the prolongation of time means that the cost of goods loss is continuously generated, and the increase of the cost of goods loss seriously affects the customer satisfaction, which in turn concerns the survival of the third party logistics enterprises.

(1) The cold chain products are characterized by continuous cargo loss costs, but since there are many types of cold chain products considered in this paper, and the calculation process is too complicated because of the different cargo loss coefficients of different products, the cargo loss costs are not considered. In the future research, complex cargo loss coefficients can be introduced into the mathematical model, which can further improve the optimization accuracy.

(2) The development of technology and the improvement of the navigation software is constantly optimized, the planning of vehicle travel distance and time is increasingly accurate, and the planning process integrates various different road conditions, especially traffic congestion. In future research, the actual planned path length and time of the navigation software can be quantified and the traffic congestion coefficient can be designed. The daily traffic congestion coefficients of different customer-to-customer road sections are then recorded, and the traffic congestion coefficients for the next transportation cycle are predicted using a time series and neural networks. Finally, the traffic congestion coefficient is added to the mathematical model for more accurate transportation scheme planning.

**Author Contributions:** Conceptualization, B.X. and Z.Z.; methodology, J.S.; software, R.G.; validation, B.X., J.S and Z.Z.; formal analysis, R.G.; data curation, B.X.; writing—original draft preparation, Z.Z.; writing—review and editing, J.S. All authors have read and agreed to the published version of the manuscript.

**Funding:** This research was funded by National Key Research and Development Program of China, grant number SQ2021YFC2800026 and The APC was funded by SQ2021YFC2800026.

**Institutional Review Board Statement:** Not applicable.

**Informed Consent Statement:** Not applicable.

**Data Availability Statement:** All data are available in the paper.

**Conflicts of Interest:** The authors declare that they have no conflicts of interest. The funders had no role in the design of the study; in the collection, analyses, or interpretation of data; in the writing of the manuscript; or in the decision to publish the results.

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
