# Peer review of "Research on Cold Chain Logistics Transportation Scheme under Complex Conditional Constraints"

_sustainability, doi:10.3390/su15108431_

Round 1
Reviewer 1 Report
This is a generally well-written and structured paper with sufficient theoretical background. In this paper, the authors present Research on cold chain logistics transportation scheme under complex conditional constraints. In my opinion, it could be published if the following issues are resolved.
1) Research highlights are inadequate. Please mention the intended contributions of the research in the highlights.
2) The introduction section should be written in more detail and structured way. It should contain the following- why is this study important now? What is the study background? What were the limitations in the previous relevant research which prompted this research? What are the study contributions? What are the specific objectives of the study? How this research intends to attain those objectives?
3) Figures 3 and 9 is almost unreadable and unrecognizable. Please enlarge both figures for easy visibility of the readers.
4) Results obtained from this research have not been discussed in detail. I would like to see a more detailed discussion of the results, with their interpretation and significance.
5) Contributions of this research need to be elaborated. How will this research contribute to the existing relevant literature? Will this research contribute to any policymaking? In fact, the authors should make a new section titled "Research Contribution", which should contain 3 subsections titled - Theoretical implications, managerial implications, and implications for the policymakers.
6)The writing of the paper needs a lot of improvement in terms of grammar, spelling, and presentations. The paper needs careful English polishing since there are many typos and poorly written sentences.
Some examples are as the following:
* Check the usage of the commas carefully.
* Check the articles including "a", "an" and "the".
* Check the required and unneeded blank spaces.
7)The literature review is brief. Some of the included papers could be briefly described. Also, a general overview of the topic could also be included. For instance, the following could be added:
*Designing an integrated responsive-green-cold vaccine supply chain network using Internet-of-Things: artificial intelligence-based solutions. Annals of Operations Research, 1-45.
Quality of English Language: Minor editing of English language required
Reviewer 2 Report
Thank you for providing me the opportunity to read the manuscript. I have some minor observations
1. LR is inadequate. Please add recent literature
2. Please explain the basis for such assumptions
3. Managerial implications need to be added
4. Please write a line in supporting the relevance of the paper for Sustainability in the abstract section
Please check for typos and minor grammatical corrections
Reviewer 3 Report
Based on considering the impact of manufacturers joining on the overall logistics distribution, the authors propose a two-stage partitioning strategy based on multiple distribution centers and demand splitting. The experimental results show that the mixed transportation scheme optimization model proposed in this paper can well solve the cold chain logistics transportation problem under complex conditions. Kindly refer to the following comments for improvement.
1)Include the state-of-the-art table in the literature review to show the research gap
2)Present the figures clearly
3)Include managerial and policy implications
4)Include future research of this study in the conclusion
Minor editing of English language required
Reviewer 4 Report
1 What’s the meaning of multi-center semi-open? I do not think this is a keyword.
2 Line 46 is wrong.
3 The introduction needs to be improved. Several citations are wrong. More papers need to be discussed. The list 1-5 are not true, such as there are lots of papers consider the empty return trips in VRP problems. “existing research rarely considers situations where demand can
be split and distribution includes different types of goods.” this is wrong, as I know, there are lots paper considered what you have listed. At last, when you write these conclusions, you need to give citations and tell readers how you get these conclusions.
4 Please clarify your innovations and the purpose of your paper.
5 Your paper focuses on cold chain logistics transportation, but your model is a normal classic VRP model, how to address cold chain logistics transportation? Delete cold chain logistics transportation, there are no more differences between your paper and the paper without cold chain logistics transportation.
6 There is cold chain logistics transportation cost during the delivery process, why you do not consider this part?
Seek for professional help
Round 2
Reviewer 4 Report
Thank you for responsing to comments raised primarily by reviewer 4. I think that this manuscript is worthy of publication and recommend acceptance.